# Aldehyde-alcohol dehydrogenase undergoes structural transition to form extended spirosomes for substrate channeling

Gijeong Kim [1], Jinsol Yang[2], Juwon Jang[1], Jin-Seok Choi[3], Andrew J. Roe [4], Olwyn Byron [5], Chaok Seok[2] & Ji-Joon Song [1✉]

Aldehyde-alcohol dehydrogenase (AdhE) is an enzyme responsible for converting acetyl-CoA to ethanol via acetaldehyde using NADH. AdhE is composed of two catalytic domains of aldehyde dehydrogenase (ALDH) and alcohol dehydrogenase (ADH), and forms a spirosome architecture critical for AdhE activity. Here, we present the atomic resolution (3.43 Å) cryo-EM structure of AdhE spirosomes in an extended conformation. The cryo-EM structure shows that AdhE spirosomes undergo a structural transition from compact to extended forms, which may result from cofactor binding. This transition leads to access to a substrate channel between ALDH and ADH active sites. Furthermore, prevention of this structural transition by crosslinking hampers the activity of AdhE, suggesting that the structural transition is important for AdhE activity. This work provides a mechanistic understanding of the regulation mechanisms of AdhE activity via structural transition, and a platform to modulate AdhE activity for developing antibiotics and for facilitating biofuel production.

---

[1] Department of Biological Sciences, KI for the BioCentury, Korea Advanced Institute of Science and Technology (KAIST), Daejeon 34141, Korea. [2] Department of Chemistry, Seoul National University, Seoul 08826, Korea. [3] Analysis Center for Research Advancement, KAIST, Daejeon 34141, Korea. [4] Institute of Infection, Immunity and Inflammation, University of Glasgow, Glasgow G12 8QQ Scotland, UK. [5] School of Life Sciences, University of Glasgow, Glasgow G12 8QQ Scotland, UK. ✉email: songj@kaist.ac.kr

Many cellular activities are executed by proteins, and proteins form large supramolecular complexes by either interacting with other proteins or self-assembly[1]. Many enzymes involved in metabolic pathways form self-assembled supramolecular architectures to compartmentalize specific activities within the structures, thereby enhancing enzymatic efficiency[2]. Aldehyde–alcohol dehydrogenase (AdhE) is a multifunctional enzyme containing two enzymatic domains responsible for aldehyde dehydrogenase (ALDH) and alcohol dehydrogenase (ADH) activities[3,4]. In addition, AdhE is also shown to have a pyruvate formate-lyase activity[3]. AdhE plays a key role in regulating NADH and acetyl-CoA homeostasis[5].

AdhE was shown to be a target for antibiotics as the depletion of AdhE in pathogenic bacteria results in nonfunctional flagella and reduces its virulence[6]. Previous studies also report that AdhE acts as a virulence factor in *Streptococcus pneumoniae*[7] and that AdhE is upregulated during infection in *Salmonella typhimurium*[8]. The observation that AdhE is important for virulence in several pathogens broadens its appeal as a drug target[9]. Furthermore, for industrial purpose, AdhE in fermentative bacteria is the key enzyme to produce ethanol[10].

AdhE forms long helical filaments called spirosome and this spirosome formation is critical for the AdhE activity[11]. In addition, AdhE spirosomes undergo a structural transition from a compact to an extended form, in the presence of cofactors[3,11]. We recently determined the atomic resolution cryo-EM structure of AdhE in compact spirosomes, revealing that four AdhE molecules form one helical turn and that ALDH and ADH activities are topologically separated within the spirosome structure where the ALDH catalytic site is located on the outside of spirosome while the ADH catalytic site is buried inside of the compact spirosome and not accessible by solvent[11]. This observation suggested that the conformational transition from compact to extended forms might be involved in regulating AdhE activity. However, the structural details of this structural transition remain unknown.

Here, we present the atomic resolution (3.43 Å) cryo-EM structure of AdhE in an extended spirosome form in the presence of cofactors. The cryo-EM structure shows that the AdhE spirosome undergoes substantial structural change to form a widely opened spirosome. In the extended AdhE spirosome, ADH catalytic pocket is accessible, while it is partially buried in the compact spirosome form. Structural comparison between compact and extended spirosomes suggests that the binding of the cofactors at ADH domains might induce the structural transition. Furthermore, this transition makes a substrate channel between the ALDH and ADH active sites accessible to solvent, implying that AdhE activity might be regulated through the structural transition.

This work provides a mechanistic understanding on the regulation of AdhE activity through structural transition between compact and extended spirosome forms.

## Results

### The cryo-EM structure of AdhE in an extended spirosome state.

AdhE is a bifunctional enzyme having ALDH and ADH activities[3,4]. AdhE forms into spirosome structure in vitro and in vivo, and the formation of spirosome is critical for its activity[3,11]. In addition, spirosomes undergo a structural transition from compact to extended forms in the presence of cofactors and this transition is suggested to be important for AdhE activity[3,11]. Our recent cryo-EM work on AdhE reveals the structural details of AdhE in a compact spirosome form[11]. To investigate molecular details of the AdhE in an extended spirosome form, we determined the cryo-EM structure of AdhE in an extended spirosome state. We found that AdhE stably exists as an extended spirosome

form in the presence of Zn²⁺, NAD⁺, and ethanol (Fig. 1a). AdhE in the presence of $Zn^{2+}$, $NAD^+$, and ethanol was vitrified on carbon grids and cryo-EM micrographs were collected using a Glacios 200 keV microscope with a Falcon III direct detector in an electron counting mode at the KAIST Analysis Center for Research Advancement (KARA). A total of 412,581 particles were picked and subjected to 2D and 3D classifications resulting in a 3.43 Å resolution cryo-EM map (Fig. 1, Supplementary Fig. 1 and Supplementary Table 1). The cryo-EM map clearly shows that AdhE forms a right-handed spirosome with wide helical grooves (Fig. 1c).

At 3.43 Å resolution, the cryo-EM map shows features of the most side chains (97.5%, 869 a.a. out of 891 a.a. in the AdhE monomer) and the atomic model was built using the high-resolution cryo-EM structure of the compact AdhE spirosome (Fig. 2a, b and Supplementary Fig. 2)[11]. A total of six AdhE molecules were built on the cryo-EM map. The cryo-EM structure reveals that AdhE forms a right-handed spirosome with 130 Å width, and four AdhE molecules make one helical turn with a pitch of 120 Å, which is consistent with the previous SAXS analysis on an extended AdhE spirosome[11]. In one helical turn, ALDH and ADH domains from two AdhE molecules are intertwined to form a dimer, and two dimers make a tetramer by interacting with each other via their ADH domains (Fig. 2c). In the cryo-EM structure, there are clear densities of NAD⁺ in both the ALDH and ADH domains, and Zn²⁺ in the ADH domain

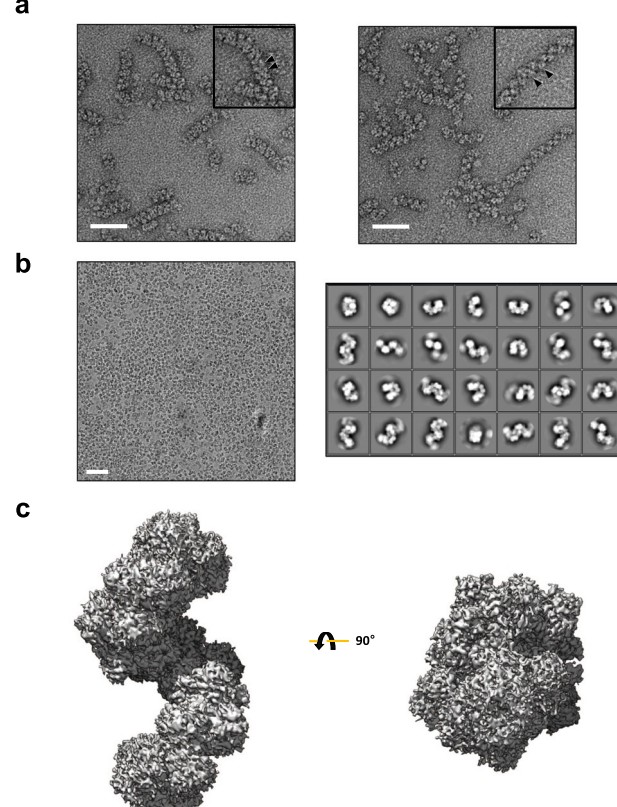

**Fig. 1 Cryo-EM analysis of AdhE spirosomes. a** Negative staining electron microscope images of AdhE spirosomes in the absence (left) and the presence (right) of the cofactors (50 μM Zn²⁺, 500 μM NAD⁺, and 10 mM ethanol). AdhE stably exists in an extended conformation in the presence of the cofactors. The scale bars show 50 nm. **b** A representative micrograph (left) and 2D class averages (right). The scale bars show 50 nm. **c** The 3.43 Å resolution cryo-EM maps of AdhE spirosomes in an extended conformation.

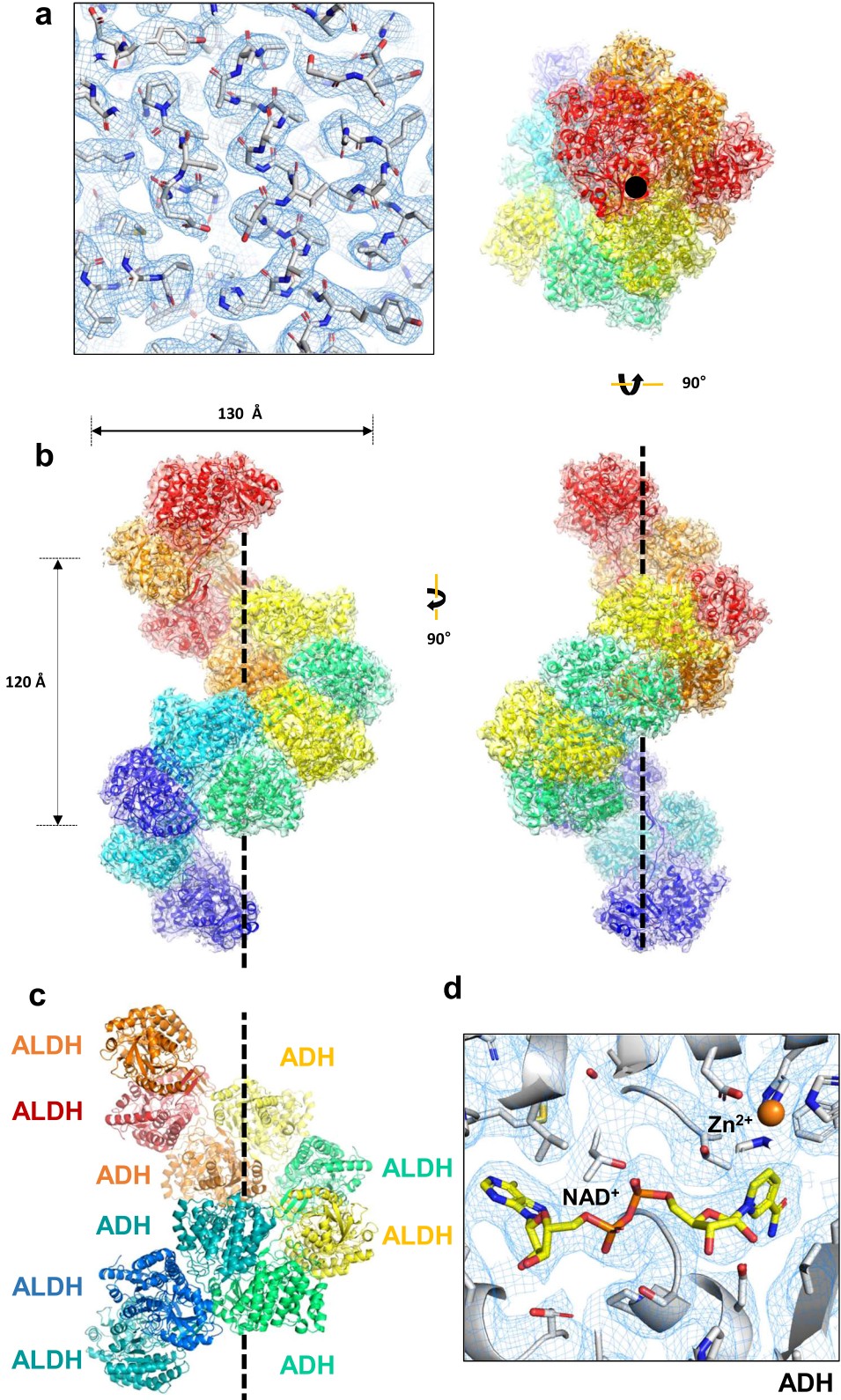

**Fig. 2 The cryo-EM structure of the AdhE spirosome in an extended conformation. a** The cryo-EM map fitted with the atomic model. **b** The atomic model of AdhE in an extended conformation in three different views. AdhE forms spirosomes with a width of 130 Å and four AdhE molecules form one helical turn with a pitch of 120 Å. AdhE molecules are labeled in different colors. **c** One helical turn with one ALDH domain at the near bottom (blue) and one ADH domain at the near top (red) is shown. One helical turn is composed of four AdhE molecules (each AdhE is colored in orange, yellow, green, and cyan). An AdhE dimer is formed by intertwining ALDH and ADH domains, and a tetramer AdhE is formed majorly by ADH–ADH interaction in a tail-to-tail manner. **d** The cryo-EM map shows clear density for $NAD^+$ and $Zn^{2+}$ near the catalytic pocket of the ADH domain.

(Fig. 2d and Supplementary Fig. 3). This high-resolution cryo-EM structure provides structural details of AdhE spirosomes in an extended form.

**Conformational transition of spirosome structures**. We recently presented the cryo-EM structure of AdhE showing that AdhE forms a compact spirosome with 70 Å helical pitch with 150 Å width, where each helical turn is mediated by inter-helical interactions between ADH domains (Fig. 3a and Supplementary Fig. 4a)[11]. The cryo-EM structure of AdhE from this present study shows that AdhE forms an extended spirosome with 120 Å helical pitch with 130 Å width (Figs. 2b and 3b). In addition, the angle of the helical turn relative to the helical axis is 40° while it is 30° in the compact spirosome (Fig. 3a, b). Furthermore, there is no inter-helical interaction in the extended AdhE spirosome (Fig. 3b and Supplementary Fig. 4b). To understand the structural details of the conformational transition from a compact to an extended spirosome, we superimposed the AdhE dimers from compact and extended spirosomes (Fig. 3c and Supplementary Movie 1). The AdhE dimer in an extended spirosome is widened by about 10° compared with the dimer in the compact spirosome. In addition, there is a 10° twist observed from the twofold axis of the AdhE dimer (Fig. 3c). These relative movements among the ALDH and ADH domains in the spirosome result in the structural transition from a compact to an extended spirosome structures (Fig. 3 and Supplementary Movie 2).

Interestingly, the inter-helical interactions in the compact AdhE spirosome occurs at the $NAD^+$ binding pockets of ADH domains and the pockets are partially buried (Fig. 3a and Supplementary Fig. 4a). In contrast, in the extended conformation where $NAD^+$ is bound, the $NAD^+$ binding pocket is widely exposed (Fig. 3b and Supplementary Fig. 4b). This observation suggests that the cofactor binding to ADH domain might play a role in inducing the conformational transition from a compact to an extended spirosome. Alternatively, the conformational transition may regulate $NAD^+$ binding and in turn, affect AdhE activity.

**A substrate channel formation in the extended AdhE spirosome**. We have shown that the spirosome formation of AdhE is required for its activity[11]. The ADH catalytic pocket located on the inside of the compact AdhE spirosome, is not readily accessible to solvent (Fig. 3a and Supplementary Fig. 4a), while the ALDH catalytic pocket located on the outside is accessible (Fig. 3b and Supplementary Fig. 4b). To understand how two catalytic activities can be combined in the AdhE spirosome structure, we examined the catalytic sites of the ALDH and ADH domains. In the extended AdhE spirosome, both the ALDH and ADH catalytic sites are readily accessible to solvent from the outside as well as the inside of the spirosome (Fig. 4a). Furthermore, the interaction between the ALDH and ADH domains of two different AdhE molecules creates a channel between the two catalytic sites, which is located at the inside of the spirosome (Fig. 4a). This channel is formed between two nicotinamide groups of $NAD^+$s bound at the ALDH and ADH domains, which is consistent with the recently published work[12]. Considering that ALDH and ADH catalytic activities mediate consecutive reactions converting acetyl-CoA to ethanol via acetaldehyde as an intermediate, it is likely that the channel observed plays a role in conveying the intermediate product. In the compact spirosome, the substrate channel still exists (Fig. 4b). However, as the access to the substrate binding site at ADH is limited due to the inter-helical interaction (Fig. 3a and Supplementary Fig. 4a), the compact spirosome is not likely an active form. To further elucidate the characteristics of the channel, we performed a docking simulation to examine whether the channel shows higher

occupancy of acetaldehyde than other parts of the protein surface. The global optimization protocol of GalaxyDock2 was used to sample the translational and rotational degrees of freedom of acetaldehyde starting from 600 randomly generated poses on the protein surface[13]. Among the final 600 poses obtained after global optimization, top 100 poses of acetaldehyde with the highest scores, as determined by GalaxyBP2score, were selected for visualization (Supplementary Movie 3)[14]. These results shows that top poses of acetaldehyde can form a continuous pathway along the channel connecting the two active sites (Fig. 4c). These results further support the idea that the spirosome formation and its structural transition is critical for AdhE activity.

**The conformational transition is critical for AdhE activity**. The cryo-EM structures of AdhE spirosomes in compact and extended forms suggest that the conformational transition from a compact to an extended form is critical for AdhE activity[11]. To investigate the role of the conformational transition, we crosslinked the spirosome to prevent the structural transition. To crosslink AdhE at the helical interface, we introduced cysteines (E817C and Q821C) at the end of the loop in the ADH domains where two ADH domains meet at the inter-helical interface (Fig. 5a). To confirm that the Cys–Cys bond is formed in this mutant, we examined the mutant AdhE spirosome using negative stained EM in the absence and the presence of dithiothreitol (DTT), a reducing agent. In the absence of DTT, the Cys mutant AdhE spirosome stays as a compact form in a condition where the wild-type (WT) AdhE forms an extended spirosome (Fig. 5b and Supplementary Fig. 5). However, in the presence of DTT reducing agents, the Cys mutant also forms an extended spirosome as the WT. In addition, the presence of crosslinked spirosome was also confirmed with SDS-PAGE analysis (Supplementary Fig. 6). The crosslinked AdhEs show smear pattern rather than clear dimer bands indicating dimer and higher oligomer states. It is possible that Cys–Cys crosslinking between ADH domains somehow could not be fully resolved as dimer even in the presence of SDS. In addition, only a small portion of AdhE was crosslinked in the SDS gel analysis. We reasoned that these results might be due that the spirosome formation and extension are dynamic processes and that we did not treat any oxidizing agent to promote Cys–Cys crosslinking not to affect the AdhE activity.

It should be noted that in the presence of DTT, the spirosome extension is less clear in both the WT and Cys mutant AdhE proteins compared with what is observed in the absence of DTT. Despite this caveat, these data suggest that the Cys mutations result in inter-helical crosslinking to prevent from forming an extended spirosome. Having the Cys mutant that cannot form an extended spirosome, we have measured the activities of the Cys mutant and WT AdhE. In the absence of DTT, the activity of the Cys mutant is substantially lower than that of the WT AdhE, while the Cys mutant and WT show similar activity in the presence of DTT, suggesting that the inter-helical crosslinking freezing the AdhE conformation in a compact form hampers the AdhE activity (Fig. 5c). Overall these data imply that the structural transition from compact to extended spirosome form is critical for AdhE activity.

## Discussion

AdhE is a bifunctional enzyme having two catalytic activities responsible for two consecutive reactions converting acetyl-CoA to ethanol[3,4]. AdhE forms into a high-order architecture, the spirosome, resulting in the membrane-less compartmentalization of two enzymatic activities[11]. Our previous cryo-EM work on AdhE shows that AdhE adopts a compact spirosome form[11]. In the compact spirosome, the two catalytic sites are topologically

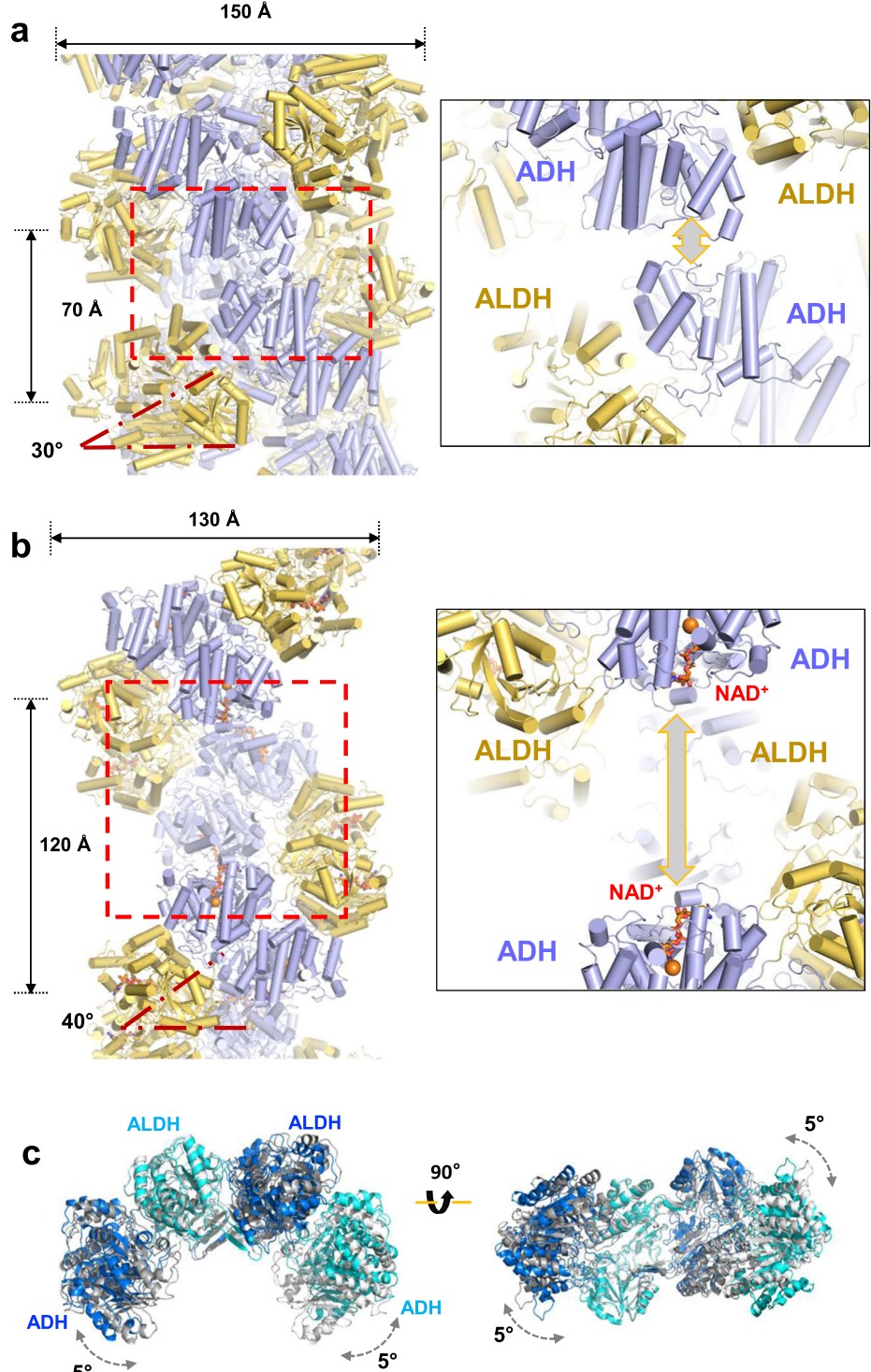

**Fig. 3 Comparison of compact and extended AdhE spirosomes. a** The cryo-EM structure of AdhE in the compact spirosome (6AHC)[11]. The helical pitch is 70 Å and the angle between the helical axis and the helical turn is 30°. The inter-helical interaction indicated with a red dotted box is zoomed in the panel (left). The ALDH domains in different helical pitches interact within the compact spirosome (shown in gray arrow). The ALDH domains are colored in yellow and the ADH domain in pale-purple. **b** The cryo-EM structure of AdhE in the extended spirosome is shown. The helical pitch is 120 Å and the angle between the helical axis and the helical turn is 40°. The ALDH domains in the extended spirosome stays far apart (left box, shown in gray arrow). **c** Superimposition of the compact (shown in gray) and extended (shown in blue and cyan) AdhE dimers. AdhE dimer in the extended spirosome is expanded and twisted by 5° at the both ends of the ADH domains compared with the compact AdhE dimer (Supplementary Movies 1 and 2).

separated, and the ADH catalytic site is not accessible by solvent as it is partially buried inside of the spirosome, while the ALDH activity resides on the outside of the spirosome[11]. The observations of two different forms of spirosome suggested that AdhE

activity might be regulated through the structural transition from compact to extended spirosomes[3,11]. Our cryo-EM structure presented here together with the compact AdhE structure clearly visualized this structural transition. In the compact AdhE

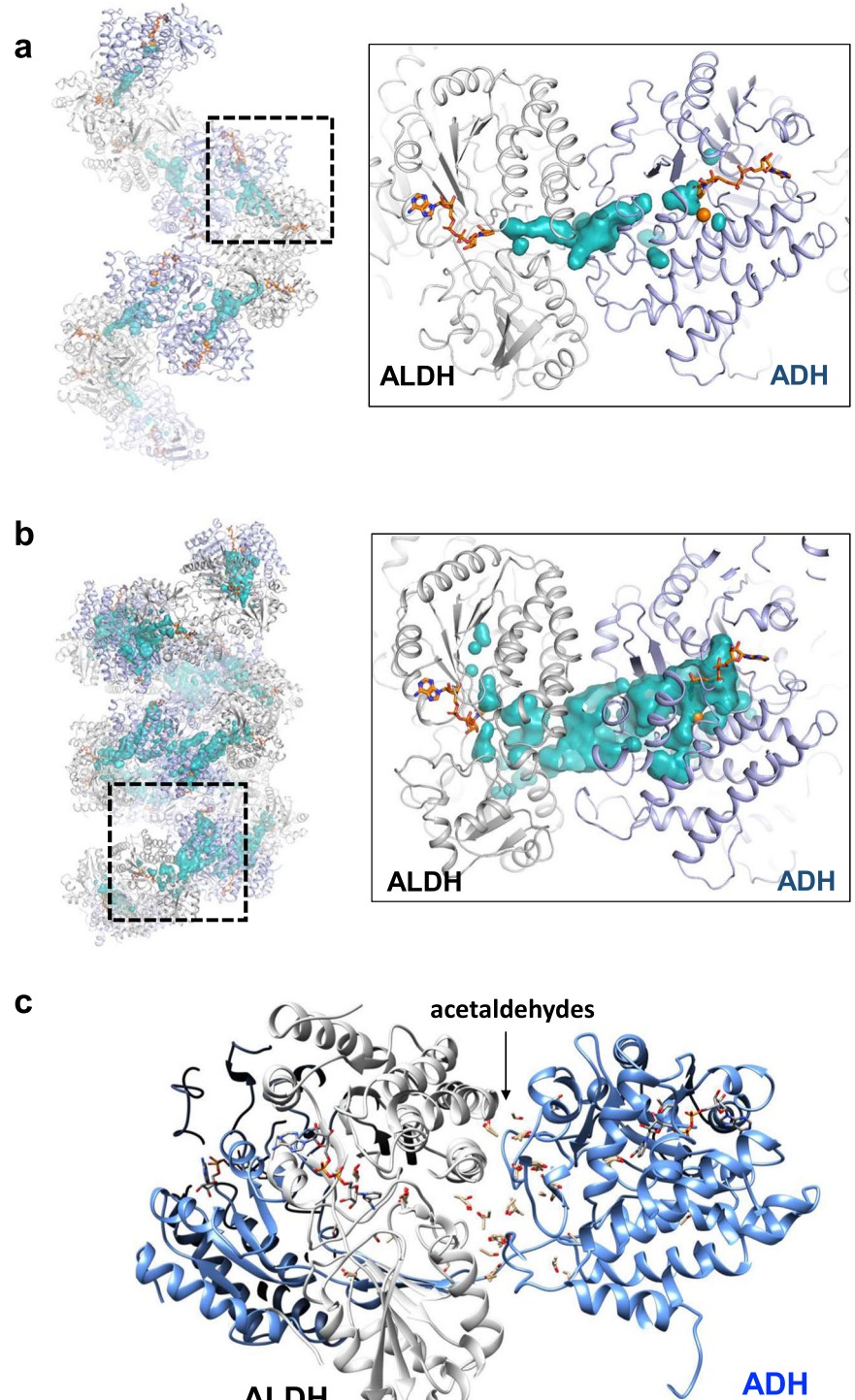

**Fig. 4 Structural transition from compact to extended spirosome makes a substrate channel accessible by solvent.** Substrate channels in the extended (**a**) and compact (**b**) AdhE spirosomes. The substrate channel between $NAD^+$s of ALDH and ADH catalytic sites from two different AdhE subunits (colored in gray and pale-purple) is shown in a cyan surface representation. $NAD^+$ in the compact spirosome (**b**) was modeled based on the obtained cryo-EM structure. **c** Molecular docking simulation of acetaldehydes shows that top 100 poses of acetaldehyde (shown in a stick model) form a continuous pathway along the substrate channel (Supplementary Movie 3).

spirosome, the cofactor binding site located at the helical interface of the spirosome is partially buried, suggesting that the cofactor binding might be involved in the structural transition (Fig. 3). Consistent with this, we found that $NAD^+$ cofactor is required for maintaining the extended AdhE spirosome form (Fig. 1b). The structural transition of spirosome seems to result from the

relative repositioning among ALDH and ADH domains within the AdhE dimer (Fig. 3c) although we cannot exclude other possibilities such as a structural transition resulting from reassembly of the AdhE protomer.

It has been intriguing why the spirosome formation is critical for AdhE activity despite that the AdhE monomer has two fully

**a**

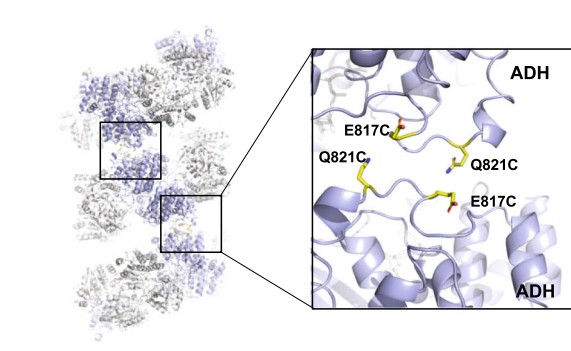

**b**

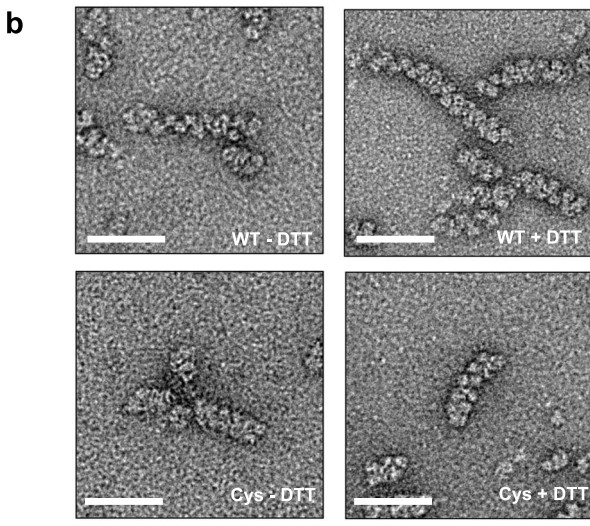

**c**

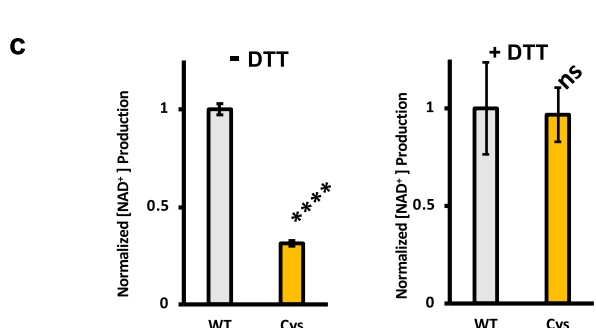

**Fig. 5 Structural transition from compacted to extended spirosome is important for AdhE activity. a** E817 and Q821 located at the inter-helical interface were mutated to cysteines to introduce cross-links to freeze the compact AdhE spirosome structure. **b** Negative stained EM analysis of the wild-type and Cys mutant AdhE spirosomes in the absence and presence of DTT. The scale bars show 50 nm. **c** A bar graph showing the relative activity of the wild-type (WT) and Cys mutant (Cys) in the absence (−DTT) and presence (+DTT) of DTT. Corresponding $p$ values are $3.06 \times 10^{-6}$ and 0.84, respectively. (****$p$ value <0.0001, NS $p$ value >0.05 from $n = 3$, two-tailed student's $t$ test).

functional intact enzymatic domains. This suggests that the reaction product of ALDH activity cannot be conveyed to an ADH domain within the monomer of AdhE. The cryo-EM structure shows that there is a channel between the ALDH and ADH catalytic sites from two different AdhE subunits. Considering that two activities are responsible for two consecutive reactions and the intermediate of these reactions is acetaldehyde, which is cytotoxic, the channel formed in the spirosome is likely

to function as a substrate channel conveying the product from the first reaction to the second (Fig. 6).

In summary, this work shows that the AdhE spirosome undergoes structural transition from compact to extended forms and imply that the AdhE activity might be regulated through this structural transition.

## Methods

**Protein purification.** The WT *adhE* gene of *Escherichia coli K12* was cloned into a pET28a vector and expressed with N-terminal 6-His tag. Protein purification was performed as described in Kim et al.[11]. The AdhE Cys mutant (E817C Q821C double mutant) was generated using a QuikChange site-directed mutagenesis kit (Stratagene) and purified in the absence of DTT.

**Negative staining electron microcopy.** Carbon coated Cu (400 mesh) grids were glow discharged for 20 s at 30 mA, using a PELCO easiGlow™ glow discharger. Three microliters of purified AdhE (0.05 mg/ml) were applied to the grid and incubated for 1 min. Excess protein drop was removed by filter paper and the grid was washed twice with distilled water and once by 1.5% (w/v) uranyl acetate, and incubated in 1.5% (w/v) uranyl acetate for 1 min. Excess uranyl acetate was removed by filter paper and the grid was dried. Negative stained protein was observed under a Tecnai F20 electron microscope (FEI) equipped with Gatan CCD camera.

**Cryo-EM image processing and structure determination.** Purified AdhE (5 mg/ml) was incubated with 50 μM ZnCl₂, 1 mM NAD⁺, and 10 mM ethanol for 30 min at room temperature. Three microliters of the cofactor bound AdhE was applied to glow-discharged R 2/2 Cu Quantifoil holey carbon grids (200 mesh). The grids were blotted for 3 s with −4 blotting force in 100% humidity at 22 °C and plunge-frozen by using a Vitrobot Mark IV (Thermo Fisher Scientific). A total of 1704 micrographs were collected using a Glacios cryo-TEM (Thermo Fisher Scientific) microscope equipped with a Falcon III direct detector in an electron counting mode. Magnification was ×92,000, 1.14 Å/pixel. The micrographs were imaged in an electron counting mode with 40 frames at 0.44 e⁻/A²/s dose rate for 92.64 s, giving a total dose of 40.76 e⁻/A². The defocus range was −0.8 to −2.8 μm with a 0.4 μm step. All the image processing was performed using RELION-3.0[15] (Supplementary Fig. 1). The movie stack was aligned with MotionCorr2[16], and CTF was corrected with CtfFinder4[17]. From 743 of selected micrographs, 412,581 particles were initially picked using template-based picking and subjected to 2D class averaging followed by initial model generation and two rounds of 3D classifications. Particle picking was performed again using a 3D reconstructed structure followed by 2D classification and 3D classification. A final 71,599 particles were used for reconstructing the 3D structure. After 3D refinement, CTF refinement, particle polishing, and post-processing, the resolution of the structure was 3.43 Å was determined from a FSC curve using a coefficient criterion of 0.143. The atomic model was generated using the published AdhE structure (6AHC), and the refinement was performed using PHENIX[18]. All figures were drawn using Chimera[19].

**Enzymatic activity assay.** To determine acetyl-CoA reductase activity of the WT and Cys mutant (E817C Q821C double mutant) AdhE, Synergy H1 microplate reader (BioTek) was used for monitoring consumption of NADH at 340 nm. The activity of the forward reaction was measured in an acetyl-CoA reductase assay mixture containing 50 mM Tris-HCl pH 8.0, 20 μM FeSO₄, 200 μM acetyl-CoA, 250 μM NADH, and 6 μg AdhE (0.06 nmol). To reduce the disulfide bridge of the AdhE Cys mutant, 10 mM DTT was added to the acetyl-CoA reductase assay mixture. To measure the activity, absorption value at 340 nm was obtained 5 min after reaction. All assays were done at 37 °C and total reaction volume was 100 μl. For SDS-PAGE analysis of AdhE Cys–Cys crosslinking, the AdhE Cys mutant was incubated in the presence and absence of 50 mM DTT at 37 °C for 30 min at 500 rpm rotation. The samples were prepared with 5X SDS buffer with/without 752 mM β-mercaptoethanol and 2 mM DTT. The crosslinking was analyzed with 8% SDS-PAGE. All enzyme assays were performed using three independent experiments (n = 3).

**Statistics and reproducibility.** All enzyme assays were performed using three independent experiments (n = 3). P values were calculated with two-tailed student's t test.

**Reporting summary.** Further information on research design is available in the Nature Research Reporting Summary linked to this article.

## Data availability

The cryo-EM map and the atomic coordinates were deposited in the EMDB and PDB (accession Code: EMD-30220 and 7BVP, respectively). Original data are available from the corresponding author upon reasonable request.

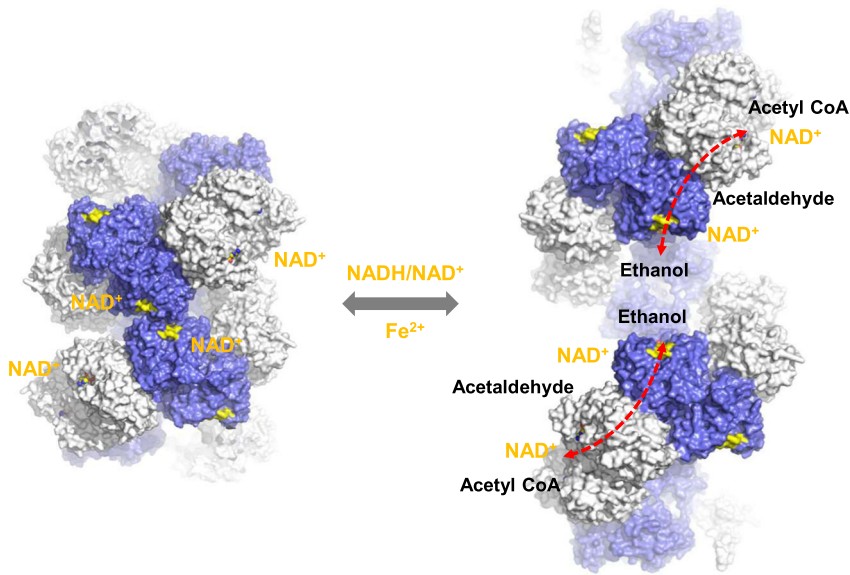

**Fig. 6 A schematic model for the conformational transition of AdhE and its implication for AdhE activity.** Conformational transition of AdhE from compact (left) to extended (right) spirosomes. The NAD+ binding pockets are highlighted in yellow. ALDH and ADH domains are shown in light gray and royal blue, respectively. Cofactor binding at the inter-helical interface in the compact spirosome might induce the formation of extended spirosomes. The substrate channels between the ALDH and ADH catalytic sites are indicated with red dotted arrows.

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

## Acknowledgements
We thank the members of the Song Lab for helpful discussions. We thank Global Science Experimental Data Hub Center (GSDC) at Korea Institute of Science and Technology Information (KISTI) for computing resources and technical support. We also thank KAIST Analysis Center for Research Advancement (KARA) for maintenance and supporting cryo-EM facility. This work is partially supported by the Grand Challenge 30 program (to J.S.) from KAIST and grants (NRF-2020R1A2B5B03001517, NRF-2016K1A1A2912057) from the National Research Foundation of Korea. This work was partially supported by the Intelligent Synthetic Biology Center (ISBC) of Global Frontier Project funded by the Ministry of Science and ICT (MSIT) (2011–003955). G.K. is a recipient of a Global Fellowship (NRF-2018H1A2A1061362).

## Author contributions
G.K., A.R., O.B., and J.S. conceived the idea. G.K., J.J., and J.-S.C. performed the cryo-EM study, and G.K. performed in vitro assay. J.Y. and C.S. examined the substrate channel with molecular docking simulation. All authors examined the data and wrote the paper.

## Competing interests
The authors declare no competing interests.
