## [Peer Review File · Communications Biology]

Reviewers' comments:

Reviewer #1 (Remarks to the Author):

In the manuscript "Aldehyde-alcohol dehydrogenase undergoes structural transition to form extended spiroosome for substrate channeling" Authors determined the structure of the AdhE spiroosomes in extended form using CryoEM. The authors showed mechanistic understanding on the regulation mechanism of AdhE activity via structural transition from the compact to extended spiroosomes caused by the substrate binding. They also found that the substrate binding induces the formation of a channel which transport the product of the ALDH domain enzymatic activity.

I have following concerns and comments about the manuscript:

1. There are only eight reference in the reference list. Authors need to list cite more reference to validate their quoted texts in the introduction. Below are some examples where the authors did not cite any reference:

..... All cellular activities are executed by proteins, and proteins form large supramolecular complexes by interacting with other proteins and self-assembly...

... AdhE plays a key role in regulating the homeostasis of the level of NADH and Acetyl-CoA.....

... We recently determined the atomic resolution cryo-EM structure of AdhE in a compact site is buried inside of the compact spiroosome and not accessible.....

Results:

2. Fig. 1a, b: Why the spiroosome lengths in negative staining and the CryoEM micrographs are different?

3. Fig. 2d: What is confidence level about the density of the NAD in the ALDH domain? Did authors check the shape of density of the NAD generated separately and tried to overlap with the density in the ALDH map?

4. Also, for the density of the Fe is very difficult to fit at this resolution. If authors really want to claim that Fe density is real, then they need to improve the resolution.

5. Fig 3b, c: what is the point of showing labels in different colors if they do not match with the corresponding map color? So, I would suggest authors to match the label and map color to each other.

6. Page 12.... Furthermore, there is no inter-helical interaction in the extended AdhE spiroosome (Fig. 3b).... Authors might need to show the expanded interface to make this point clear.

7. Fig. 3d: What is the reduction of the overall pocket size? What are the red highlighted parts of the map in Fig 3e?

8. Page 15; We have shown that the spiroosome formation ofe while ALDH catalytic pocket located at theof ALDH and ADH domains; authors need to cite the reference or point the figure showing these results.

9. From the fig 4b, it appears that authors channel is deeper as compared to width, but authors claimed in opposite way, why? (The dimensions of the channel are about 10-20 Å wide and 5-10 Å deep).

10. Page 17,18; Fig 5b: Authors need to test the oligomer formation induced by cys-cys bond in the non-reducing SDS-PAGE gels. Also, Authors may need to collect negative stained micrographs as high magnification to see comparatively better difference in the pitch of the spiroosomes formed in two conditions. Also, does the AdhE has any intrinsic cysteine? If yes does the authors, make it cys-less before introducing the cys residues at the ADH interface if not they might need to test the as well?

11. I would also suggest the authors to get the high resolution structure using higher end microscopes and that will make conclusion very strong.

Reviewer #2 (Remarks to the Author):

Overall appraisal:-

This manuscript describes the 4.80 Angstrom cryo-EM structure of the AdhE spiroosome comprised of aldehyde dehydrogenase (ALDH) and alcohol dehydrogenase (ADH) domains. The study shows the AdhE spiroosome can exist in a compact and extended conformation that facilitates substrate channel formation important for catalytic function, as demonstrated by cryo-EM and enzyme activity studies of wild-type and cys (disulphide trap) mutants in the absence and presence of reducing agent. The experimental design and conclusions drawn for most parts are relevant, but there are some points requiring revision and/or clarification as documented below. Accordingly, I cannot recommend the manuscript for publication in Communications Biology in its current form.

General Comments for Revision:-

- (1) All statements describing facts based on previous observations should cite relevant publication(s). For example, page 3, the first and last sentences of the first paragraph of the Introduction and the last three sentences of the second paragraph of the Introduction, should be supported by references. Likewise, the statements in the first sentences under the relevant headings on pages 5, 15 and 20 should also be supported by reference(s).
- (2) Some sentences are poorly written and therefore require correction to better communicate the findings of the study. For example, page 5 (twelfth line of the first paragraph) should read as 'A total of 264,787...' (rather than 'Total 264,787...'); page 5 (third line of the second paragraph) should read as 'A total of four....' (rather than 'Total four...'); and page 5 (fifth line of the second paragraph) should read as '...forms a right-handed spiroosome with a width of 100 angstrom...' (rather than '...form a right-handed spiroosome with 100 angstrom width...'). Proof and correct elsewhere in the manuscript (where applicable).
- (3) There are a number of extra spaces in the text throughout the manuscript. For example, page 4, 2nd last line between the terms 'AdhE' and 'activity'; page 15, second sentence under heading 'The formation of a substrate channel in the AdhE extended spiroosome' between the term 'solvents' and the full stop at the end of the sentence; page 15, ninth line under heading 'The formation of a substrate channel in the AdhE extended spiroosome' between the terms 'a' and 'channel'; page 17, line 13 under the heading 'Conformational transition from compact to extended spiroosome is critical for (of) activity' between the terms 'caveat,' and 'these'; and page 21, second last line between the terms 'that' and 'the'. Check elsewhere in the manuscript and correct (where applicable).

Specific Points for Revision:-

- (4) Page 5: the terms 'in-vitro' and 'in-vivo' are not presented conventionally – do not need to be hyphenated but should be in italics.

- (5) Page 5, 2nd paragraph RE: sentence ‘...shows features of the most side chains...’: The authors should state the number or percentage of (i.e. quantify) side chains observed in the cryo-EM structure rather than describe qualitatively as ‘most’.
- (6) Page 5/ Fig. 2(d): The last sentence of the Results on the bottom of page 5 contradicts the legend for Fig. 2(d). Page 5 indicates that the cofactors are bound ‘to only ADH domains’, but the figure legend indicates the cofactors are ‘near the ALDH pocket’. The authors need to correct this contradiction.
- (7) Fig 2(d): The label for ‘Zn²⁺’ in panel d should be moved closer to the actual bound Zn²⁺ (consistent with the label for ‘NAD⁺’).
- (8) Page 15, final line: the term ‘implying’ should be replaced to ‘providing further support’ (i.e. in addition to studies described in reference 6).
- (9) Page 17, top: change ‘of’ to ‘for’ in the heading to read as ‘Conformational transition from compact to extended spiroosome is critical for activity’
- (10) Fig. 5(c): bar graphs with error bars are shown for the enzymatic activity of the WT and Cys mutants of AdhE, and P-values are stated in the legend. However, the figure legend or the relevant text in the Results and Methods sections do not state the number of experiments (N), number of technical replicates (n), and the nature of statistical analysis (e.g. student’s t-test, bootstrapping, etc.) performed for these enzyme assays. The authors must clarify how P-values were determined, and indeed, describe the experimental design so that one can appreciate if the P-values reported are relevant for the data shown.

Reviewer #3 (Remarks to the Author):

Kim et al. solved the structure of an extended form of Aldehyde-alcohol dehydrogenase spiroosome when incubating with cofactors. The same author has previously published a compact form of AdhE spiroosome. Combining the two structures, the authors show that the spiroosome undergoes a transition from compact to extended form. This structural transition leads to the expose of a previously buried ADH catalytic site and also to the formation of a substrate channel that is possibly used to transport the reaction intermediate to the next catalytic site. Since this channel is formed between two AdhE dimers, only existed in the extended form of spiroosome, the authors suggest that the AdhE activity might be regulated by the structural transition. The manuscript is well presented and straightforward to read. However, the impact is limited by the fact that due to the limited resolution, critical mechanistic knowledge regarding how the cofactor binding triggers the structural transition that regulates the catalytic activity is missing.

Major points:

- 1). Although the authors present a compact and an extended form of AdhE spiroosome, it is not clear that if the structure directly transforms from compact form to extended form. One other possibility is that the compact form spiroosome first disassemble into protomers and reassemble into extended form.
- 3). Due to the lower resolution, the structural analysis of the interaction between cofactor and AdhE is not as thorough as one would wish. Also, it is hard to assess the quality of the cryo-EM map with current figures. The authors should include figures of a few detailed density maps of secondary structures with atomic model overlay.
- 4). One outstanding feature of this manuscript is the formation of a substrate channel in the extended spiroosome. The authors speculate that the channel plays a role in conveying the intermediate product. It will be really exciting if the authors could perform molecular dynamic simulations to illustrate that acetaldehyde can be transport in the channel.

5). The structural transition is very interesting. However, it seems the author suggests there are only two forms of spiroosome. I am wondering if there are other intermediate states between these two forms.

Adding supplementary movies of the structural transition will help the reader understand the process better.

Minor points:

1). Have the authors try incubating AdhE with Zn, NAD, and CoA. Will spiroosome with CoA be the same as the reported structure.

2). Fig. 3c not very helpful in terms of help the reader understand the difference between compact and extended AdhE dimers. A supplementary movie shows a morph of two structures, and how the dimers are positioned in the spiroosome will be helpful.

3). In Fig. 5, the cysteine crosslinking should be validated by an SDS PAGE.

Reponses to the reviewers' comments

Reviewers' comments:

Reviewer #1 (Remarks to the Author):

In the manuscript “Aldehyde-alcohol dehydrogenase undergoes structural transition to form extended spiroosome for substrate channeling” Authors determined the structure of the AdhE spiroosomes in extended form using CryoEM. The authors showed mechanistic understanding on the regulation mechanism of AdhE activity via structural transition from the compact to extended spiroosomes caused by the substrate binding. They also found that the substrate binding induces the formation of a channel which transport the product of the ALDH domain enzymatic activity.

I have following concerns and comments about the manuscript:

1. There are only eight reference in the reference list. Authors need to list cite more reference to validate their quoted texts in the introduction. Below are some examples where the authors did not cite any reference:

..... All cellular activities are executed by proteins, and proteins form large supramolecular complexes by interacting with other proteins and self-assembly...

.... AdhE plays a key role in regulating the homeostasis of the level of NADH and Acetyl-CoA.....

.... We recently determined the atomic resolution cryo-EM structure of AdhE in a compact
..... site is buried inside of the compact spiroosome and not accessible.....

→ We apologize for this. In the revised manuscript, we cited proper references to validate our quotes in the introduction.

Results:

2. Fig. 1a, b: Why the spiroosome lengths in negative staining and the CryoEM micrographs are different?

→ When AdhE is purified, AdhE shows diverse lengths of the spiroosome. The negative staining was done from the sample having diverse lengths of the spiroosome. For cryo-EM data collection, we purposely choose a fraction from gel-filtration containing 2-3 turns of the spiroosome. Therefore, we expect the spiroosome lengths in negative staining and cryo-EM micrographs differ.

3. Fig. 2d: What is confidence level about the density of the NAD in the ALDH domain? Did authors check the shape of density of the NAD generated separately and tried to overlap with the density in the ALDH map?

→ In the revised manuscript, we present a new atomic resolution (3.43 Å) resolution cryo-EM structure from a new data set. The new cryo-EM map unambiguously resolved the densities for NAD⁺s in both ALDH and ADH domains (Figure 2 and Supplementary Fig. 3.)

4. Also, for the density of the Fe is very difficult to fit at this resolution. If authors really want to claim that Fe density is real, then they need to improve the resolution.

→ During the sample preparation, we used Zn²⁺ instead of Fe and we found that Zn²⁺ stabilizes the extended conformation. Therefore, we assumed that the density is Zn²⁺. In the revised manuscript, as mentioned above, we improved the resolution and included the cryo-EM map near Zn²⁺ (Figure 2 and Supplementary Fig. 3.). The new high-resolution structure clearly resolved Zn²⁺ density.

5. Fig 3b, c: what is the point of showing labels in different colors if they do not match with the corresponding map color? So, I would suggest authors to match the label and map color to each other.

→ We apologized for not being consistent in the color scheme. We made new figures accordingly.

6. Page 12.... Furthermore, there is no inter-helical interaction in the extended AdhE spiroosome (Fig. 3b).... Authors might need to show the expanded interface to make this point clear.

→ The comparison between the compact AdhE spiroosome (Fig. 3a) and extended AdhE spiroosome (Fig. 3b) clearly shows that there is no-inter helical interaction in the extended AdhE spiroosome. In the revised manuscript, we included figures showing the expanded interfaces for the extended spiroosome form as well as the compact spiroosome form as Supplementary Fig. 4.

7.Fig.3d: What is the reduction of the overall pocket size? What are the red highlighted parts of the map in Fig 3e?

→ We apologize for not being clear for this. We did not observe the reduction of the pocket size in the compact form, compared with the extended form. In the revised manuscript, we revised the entire Fig. 3. with the new atomic resolution cryo-EM structure.

8.Page 15; We have shown that the spiroosome formation ofe while ALDH catalytic pocket located at theof ALDH and ADH domains; authors need to cite the reference or point the figure showing these results.

→ We apologize for these. We cited a reference and referred a figure in the revised manuscript.

9.From the fig 4b, it appears that authors channel is deeper as compared to width, but authors claimed in opposite way, why? (The dimensions of the channel are about 10-20 Å wide and 5-10 Å deep).

→ In this revised manuscript, we obtained a new cryo-EM structure of AdhE in atomic-resolution (3.43 Å), which clearly resolve 98% of amino acids (869 a.a. out of 891 a.a. in AdhE monomer). Based on this high-resolution structure, we redescribed the substrate channel (Fig. 4 and Fig. 6), which is also consistent with the recently published work by Pony et al., Nat. Commun. (2020).

10.Page 17,18; Fig 5b: Authors need to test the oligomer formation induced by cys-cys bond in the non-reducing SDS-PAGE gels. Also, Authors may need to collect negative stained micrographs as high magnification to see comparatively better difference in the pitch of the spiroosomes formed in two conditions. Also, does the AdhE has any intrinsic cysteine? If yes does the authors, make it cys-less before introducing the cys residues at the ADH interface if not they might need to test the as well?

→ We have analyzed the Cys-Cys crosslinking of AdhE by SDS-PAGE included as Supplementary Fig. 6. In this analysis, the crosslinked AdhEs show smear pattern rather than a clear dimer. Due to the nature of spiroosome, it is possible that Cys-Cys crosslinking between AHD domains somehow could not be fully resolved as dimer even in the presence of SDS. In addition, we observed that only small portion of AdhE were crosslinked. We reasoned that these results might be due that the spiroosome formation and expansion are dynamic processes and that we did not treat any oxidizing agent to promote Cys-Cys crosslinking not to affect the AdhE activity.

Regarding with the negative stained micrographs, as the current micrographs were imaged at sufficiently high-magnification, we instead enlarged Fig. 5b for better visualization and included the original wide-view images as Supplementary Fig. 5.

AdhE has 9 intrinsic cysteines. Among these, several cysteines near the NAD⁺ binding pocket in ALDH domain are shown to be critical for ALDHase activity (Tsybovsky and Krupenko, J. Biol. Chem. 2011). Therefore, it is not practically possible to make cys-less AdhE without affecting its activities. Despite the presence of the intrinsic cysteines, the activity of AdhE cys mutant having additional cysteine residues at the ADH interface depends on DTT, while wild-type AdhE activity is not dependent on the presence of DTT. Therefore, we reason that the activity difference between wild-type and Cys mutant is due to the introducing cysteine residues at the helical-interface, and that having intrinsic cysteines does not affect our conclusion.

11.I would also suggest the authors to get the high resolution structure using higher end microscopes and that will make conclusion very strong.

→ We thank for the reviewer's suggestion. We indeed collected new data and present a new cryo-EM structure in 3.43 Å resolution in the revised manuscript.

Reviewer #2 (Remarks to the Author):

Overall appraisal:-

This manuscript describes the 4.80 Angstrom cryo-EM structure of the AdhE spiroosome comprised of aldehyde dehydrogenase (ALDH) and alcohol dehydrogenase (ADH) domains. The study shows the AdhE spiroosome can exist in a compact and extended conformation that facilitates substrate channel formation important for catalytic function, as demonstrated by cryo-EM and enzyme activity studies of wild-type and cys (disulphide trap) mutants in the absence and presence of reducing agent. The experimental design and conclusions drawn for most parts are relevant, but there are some points requiring revision and/or clarification as documented below. Accordingly, I cannot recommend the manuscript for publication in Communications Biology in its current form.

General Comments for Revision:-

(1) All statements describing facts based on previous observations should cite relevant publication(s). For example, page 3, the first and last sentences of the first paragraph of the Introduction and the last three sentences of the second paragraph of the Introduction, should be supported by references. Likewise, the statements in the first sentences under the relevant headings on pages 5, 15 and 20 should also be supported by reference(s).

→ We apologize for not citing references. We cited references when we mentioned the previous results.

(2) Some sentences are poorly written and therefore require correction to better communicate the findings of the study. For example, page 5 (twelfth line of the first paragraph) should read as 'A total of 264,787...' (rather than 'Total 264,787...'); page 5 (third line of the second paragraph) should read as 'A total of four....' (rather than 'Total four...'); and page 5 (fifth line of the second paragraph) should read as '...forms a right-handed spiroosome with a width of 100 angstrom...' (rather than '...form a right-handed spiroosome with 100 angstrom width...'). Proof and correct elsewhere in the manuscript (where applicable).

→ We appreciate the reviewer's corrections. We carefully proofread throughout the revised manuscript.

(3) There are a number of extra spaces in the text throughout the manuscript. For example, page 4, 2nd last line between the terms 'AdhE' and 'activity'; page 15, second sentence under heading 'The formation of a substrate channel in the AdhE extended spiroosome' between the term 'solvents' and the full stop at the end of the sentence; page 15, ninth line under heading 'The formation of a substrate channel in the AdhE extended spiroosome' between the terms 'a' and 'channel'; page 17, line 13 under the heading 'Conformational transition from compact to extended spiroosome is critical for (of) activity' between the terms 'caveat,' and 'these'; and page 21, second last line between the terms 'that' and 'the'. Check elsewhere in the manuscript and correct (where applicable).

→ We appreciate the reviewer's corrections. We carefully proofread throughout the revised manuscript.

Specific Points for Revision:-

(4) Page 5: the terms 'in-vitro' and 'in-vivo' are not presented conventionally – do not need to be hyphenated but should be in italics.

→ We revised accordingly.

(5) Page 5, 2nd paragraph RE: sentence '...shows features of the most side chains...': The authors should state the number or percentage of (i.e. quantify) side chains observed in the cryo-EM structure rather than describe qualitatively as 'most'.

→ The new high-resolution structure resolved 869 a.a. out of 891 a.a. in the AdhE monomer. We added an explanation as "...the cryo-EM map shows features of the most side chains (97.5%, 869 a.a. out of 891 a.a. in the AdhE monomer) and the atomic model was built using the high-resolution cryo-EM structure of the compact AdhE spiroosome.."

(6) Page 5/ Fig. 2(d): The last sentence of the Results on the bottom of page 5 contradicts the legend for Fig. 2(d). Page 5 indicates that the cofactors are bound 'to only ADH domains', but the figure legend indicates the cofactors are 'near the ALDH pocket'. The authors need to correct this contradiction.

→ We sorry for the mistake. We corrected accordingly. In the revised manuscript, we included the high-resolution (3.4Å) cryo-EM structure of AdhE from new data collected during the revision. In the new high-resolution structure, we were able to see clear densities for NAD⁺s in both ALDH and ADH domain. We included the cryo-EM maps near the NAD⁺s and Zn²⁺ as Fig. 2d and Supplementary Fig. 3.

(7) Fig 2(d): The label for 'Zn2+' in panel d should be moved closer to the actual bound Zn2+ (consistent with the label for 'NAD+').

→ As we obtained the high-resolution structure during revision, we redrew the figure and followed the reviewer's suggestion.

(8) Page 15, final line: the term 'implying' should be replaced to 'providing further support' (i.e. in addition to studies described in reference 6).

→ We revised accordingly.

(9) Page 17, top: change 'of' to 'for' in the heading to read as 'Conformational transition from compact to extended spiroosome is critical for activity'

→ We revised accordingly.

(10) Fig. 5(c): bar graphs with error bars are shown for the enzymatic activity of the WT and Cys mutants of AdhE, and P-values are stated in the legend. However, the figure legend or the relevant text in the Results and Methods sections do not state the number of experiments (N), number of technical replicates (n), and the nature of statistical analysis (e.g. student's t-test, bootstrapping, etc.) performed for these enzyme assays. The authors must clarify how P-values were determined, and indeed, describe the experimental design so that one can appreciate if the P-values reported are relevant for the data shown.

→ We performed three independent experiments for measuring the activity, and calculated p-values by student's t-test. We included this information in the figure legends as well and methods in the revised manuscript.

Reviewer #3 (Remarks to the Author):

Kim et al. solved the structure of an extended form of Aldehyde-alcohol dehydrogenase spiroosome when incubating with cofactors. The same author has previously published a compact form of AdhE spiroosome. Combining the two structures, the authors show that the spiroosome undergoes a transition from compact to extended form. This structural transition leads to the expose of a previously buried ADH catalytic site and also to the formation of a substrate channel that is possibly used to transport the reaction intermediate to the next catalytic site. Since this channel is formed between two AdhE dimers, only existed in the extended form of spiroosome, the authors suggest that the AdhE activity might be regulated by the structural transition. The manuscript is well presented and straightforward to read. However, the impact is limited by the fact that due to the limited resolution, critical mechanistic knowledge regarding how the cofactor binding triggers the structural transition that regulates the catalytic activity is missing.

Major points:

1). Although the authors present a compact and an extended form of AdhE spiroosome, it is not clear that if the structure directly transforms from compact form to extended form. One other possibility is that the compact form spiroosome first disassemble into protomers and reassemble into extended form.

→ The structural transition from compact to extended forms does not seem to require substantial structural rearrangements in AdhE monomer or dimer as there are relative domain movements shown in Fig. 3c. Although we cannot exclude a possibility that the compact form spiroosome first disassemble into protomers and reassemble into extended form, this strategy would be energetically expensive as the hydrophobic interactions among AdhE have been disrupted. Regarding the reviewer's comments, we included a sentence "*....although we cannot exclude other possibilities such as the structural transition resulted from reassembly of the AdhE protomer.*" in the discussion.

3). Due to the lower resolution, the structural analysis of the interaction between cofactor and AdhE is not as thorough as one would wish. Also, it is hard to assess the quality of the cryo-EM map with current figures. The authors should include figures of a few detailed density maps of secondary structures with atomic model overlay.

→ In the revised manuscript, we present a new high-resolution (3.43Å) structure from new data. The high-resolution structure clearly resolved the side-chains (97.5%, 869 a.a. out of 891 a.a. in the AdhE monomer) and the cofactors as shown in Fig. 2a and Supplementary Fig. 3. According to the reviewer's suggestion, we included several detailed density maps of secondary structures with atomic model overlay as Supplementary Fig. 2.

4). One outstanding feature of this manuscript is the formation of a substrate channel in the extended spiroosome. The authors speculate that the channel plays a role in conveying the intermediate product. It will be really exciting if the authors could perform molecular dynamic simulations to illustrate that acetaldehyde can be transport in the channel.

→ With the new high-resolution structure, we reinterpret the channel and performed molecular docking simulation. The docking simulation was performed to check whether the channel shows higher occupancy of acetaldehyde than other parts of the protein surface. The global optimization protocol of GalaxyDock2 was used to sample the translational and rotational degrees of freedom of acetaldehyde starting from 600 randomly generated poses on the protein surface. Among the final 600 poses obtained after global optimization, top 100 poses of acetaldehyde with the highest scores by GalaxyBP2score was selected for visualization. It can be seen from the figure that top poses of acetaldehyde can form a continuous pathway along the channel connecting the two active sites. We included Fig. 4 and described these results: *“To further examine the characters of the channel, we performed a docking simulation to examine whether the channel shows higher occupancy of acetaldehyde than other parts of the protein surface. The global optimization protocol of GalaxyDock2 was used to sample the translational and rotational degrees of freedom of acetaldehyde starting from 600 randomly generated poses on the protein surface⁹. Among the final 600 poses obtained after global optimization, top 100 poses of acetaldehyde with the highest scores by GalaxyBP2score was selected for visualization¹⁰. It can be seen from the figure that top poses of acetaldehyde can form a continuous pathway along the channel connecting the two active sites (Fig. 4c).”*

5). The structural transition is very interesting. However, it seems the author suggests there are only two forms of spiroosome. I am wondering if there are other intermediate states between these two forms.

Adding supplementary movies of the structural transition will help the reader understand the process better.

→ According to the reviewer’s suggestion, we include two morphing movies of structures from the compact to the extended forms in a dimer and a spiroosome as a Supplementary Movie 1 and 2.

Minor points:

1). Have the authors try incubating AdhE with Zn, NAD, and CoA. Will spiroosome with CoA be the same as the reported structure.

→ Although we did not try the condition in the presence of CoA, the recent paper, which was published by Pony et al., Nat. Comms (2020) during the revision determined the cryo-EM structure of AdhE in the extended form in the presence of all cofactors including CoA, indicating that the structure is same in the presence of CoA.

2). Fig. 3c not very helpful in terms of help the reader understand the difference between compact and extended AdhE dimers. A supplementary movie shows a morph of two structures, and how the dimers are positioned in the spiroosome will be helpful.

→ According to the reviewer’s suggestion, we include a morphing movie of two structures as a Supplementary Movie 2.

3). In Fig. 5, the cysteine crosslinking should be validated by an SDS PAGE.

→ We have analyzed the Cys-Cys crosslinking of AdhE by SDS-PAGE included as Supplementary Fig. 6. In this analysis, the crosslinked AdhEs show smear pattern rather a clear dimer and only small portion of AdhE were crosslinked. We reasoned that these results might be due that the spiroosome formation and expansion are dynamic processes and that we did not treat any oxidizing agent to promote Cys-Cys crosslinking not to affect the AdhE activity. In addition, due to the nature of spiroosome, it is possible that Cys-Cys crosslinking between AHD domains somehow could not be fully resolved as dimer even in the presence of SDS.

In the revised manuscript, we added sentences *“In addition, the presence of crosslinked spiroosome was also conformed with SDS-PAGE analysis (Supplementary Fig. 6). The crosslinked AdhEs show smear pattern rather than clear dimer bands indicating dimer and higher oligomer states. It is possible that Cys-Cys crosslinking between AHD domains somehow could not be fully resolved as dimer even in the presence of SDS. In addition, only a small portion of AdhE was crosslinked in the SDS gel analysis. We reasoned that these results might be due that the spiroosome formation and expansion are dynamic processes and that we did not treat any oxidizing agent to promote Cys-Cys crosslinking not to affect the AdhE activity.”*

REVIEWERS' COMMENTS:

Reviewer #1 (Remarks to the Author):

In the manuscript "Aldehyde-alcohol dehydrogenase undergoes structural transition to form extended spiroosome for substrate channeling" Authors determined the structure of the AdhE spiroosomes in extended form using CryoEM and showed that channel formed between the two AdhE dimers might be used for the transportation of the enzymatic activity products of AdhE. The manuscript is now in better shape as compared to the original manuscript. They also updated the structure with improved resolution, that makes their interpretations more reliable. Though there are some spots where the manuscript can be improved such as cross-linking data, increasing resolution by using high end microscopes and overall clashscore of the model, but that do not restrict the importance of the manuscript and worth to publish. The authors responded to all the comments and concerns raised by the reviewers, but still the number of references cited are enough. For example, there plenty of the literature available for the assembly of proteins, but authors just cited one reference that is not fair to the other scientist who put lots of efforts to establish it. Here are some more suggestions for authors:

1. The authors did not mention how they prepared figures for the publication and not cited them anywhere in the manuscript.
2. FSC curve in the supplementary figure 1, should have curves for unmasked, masked, randomized, and corrected FSCs. Also, it is better to report resolution at 0.5 FSC cut off, which is accepted in the field these days.
3. Scale for the resmap, in the supplementary figure 1, need more scale points.
4. The current structure is from the 22.9% of the particles, does authors refined the other 77 % of the particles to test the conformation of the spiroosomes? The authors might get intermediate states. If that is the case, authors might be able to explain the complete mechanism of the product translation through the channel.
5. Authors might also need to show the movie of the molecular dynamics simulations performed.

Reviewer #2 (Remarks to the Author):

The authors have adequately addressed my concerns with the original submission. However, the authors should add the following sentence in the Methods under the heading 'Enzymatic Assay': 'All enzyme assays were performed using three independent experiments (n = 3).' This is important to mention, since the authors have appropriately calculated P values from 3 independent experiments (not 3 technical replicates of the same experiment, which some confuse as n = 3).

Reviewer #3 (Remarks to the Author):

The revised manuscript has vastly improved, especially with the higher resolution cryo-EM map. The authors have addressed my concerns.

REVIEWERS' COMMENTS:

Reviewer #1 (Remarks to the Author):

In the manuscript “Aldehyde-alcohol dehydrogenase undergoes structural transition to form extended spiroosome for substrate channeling” Authors determined the structure of the AdhE spiroosomes in extended form using CryoEM and showed that channel formed between the two AdhE dimers might be used for the transportation of the enzymatic activity products of AdhE. The manuscript is now in better shape as compared to the original manuscript. They also updated the structure with improved resolution, that makes their interpretations more reliable. Though there are some spots where the manuscript can be improved such as cross-linking data, increasing resolution by using high end microscopes and overall clashscore of the model, but that do not restrict the importance of the manuscript and worth to publish. The authors responded to all the comments and concerns raised by the reviewers, but still the number of references cited are enough. For example, there plenty of the literature available for the assembly of proteins, but authors just cited one reference that is not fair to the other scientist who put lots of efforts to establish it.

Here are some more suggestions for authors:

1. The authors did not mention how they prepared figures for the publication and not cited them anywhere in the manuscript.

→ We thank the reviewer for the comments. All the figures were drawn using Chimera. We cited a proper reference for Chimera and mentioned at the method section as “All figures were drawn using Chimera¹⁹.”

2. FSC curve in the supplementary figure 1, should have curves for unmasked, masked, randomized, and corrected FSCs. Also, it is better to report resolution at 0.5 FSC cut off, which is accepted in the field these days.

→ According to the reviewer’s suggestion, we revised the Supplementary Fig. 1 to show FSC curves for unmasked, masked, randomized and corrected FSC. In addition, we used 0.5 FSC cut off, which gives 3.9 Å resolution. We included both criteria in Supplementary Figure 1.

3. Scale for the resmap, in the supplementary figure 1, need more scale points.

→ According to the reviewer’s suggestion, we revised the resmap in the Supplementary Fig. 1.

4. The current structure is from the 22.9% of the particles, does authors refined the other 77 % of the particles to test the conformation of the spiroosomes? The authors might get intermediate states. If that is the case, authors might be able to explain the complete mechanism of the product translation through the channel.

→ I appreciate the reviewer’s comment on this. Although it is possible that the other 77% of the particles might contain intermediate states. However, further refinements with the 77% particles did not generate a high resolution structure, which might be the mixture of several states.

5. Authors might also need to show the movie of the molecular dynamics simulations performed.

→ We thank the reviewer for the comment. We include a movie for the molecular docking simulations as Supplementary video 3.

Reviewer #2 (Remarks to the Author):

The authors have adequately addressed my concerns with the original submission. However, the authors should add the following sentence in the Methods under the heading 'Enzymatic Assay': ‘All enzyme assays were performed using three independent experiments (n = 3).’ This is important to mention, since the authors have appropriately calculated P values from 3 independent experiments (not 3 technical

replicates of the same experiment, which some confuse as $n = 3$).

→ We thank Reviewer#2 for the comment. According to the reviewer's suggestion, we include a sentence in the method as 'All enzyme assays were performed using three independent experiments (n=3).'

Reviewer #3 (Remarks to the Author):

The revised manuscript has vastly improved, especially with the higher resolution cryo-EM map. The authors have addressed my concerns.

→ We thank Reviewer#3 for all the constructive comments.